# Schistosoma and Other Relevant Helminth Infections in HIV-Positive Individuals—An Overview

**DOI:** 10.3390/tropicalmed4020065

**Published:** 2019-04-12

**Authors:** Amrei von Braun, Henning Trawinski, Sebastian Wendt, Christoph Lübbert

**Affiliations:** 1Division of Infectious Diseases and Tropical Medicine, Leipzig University Hospital, University of Leipzig, 04103 Leipzig, Germany; henning.trawinski@medizin.uni-leipzig.de (H.T.); christoph.luebbert@medizin.uni-leipzig.de (C.L.); 2Interdisciplinary Center for Infectious Diseases, Leipzig University Hospital, 04103 Leipzig, Germany; sebastian.wendt@medizin.uni-leipzig.de; 3Institute for Medical Microbiology and Epidemiology of Infectious Diseases, Leipzig University Hospital, 04103 Leipzig, Germany

**Keywords:** HIV, helminths, schistosomiasis, neglected tropical diseases

## Abstract

For many years, researchers have postulated that helminthic infections may increase susceptibility to HIV, and that immune activation may have contributed to the extensive spread of HIV in sub-Saharan Africa. In the meantime, immunological studies have provided some evidence in support of this hypothesis, while cross-sectional clinical studies were able to further support the assumed association between HIV infection and selected helminthic co-infections. However, as many of the helminthic infections relevant to HIV-infected patients belong to the group of “neglected tropical diseases”, as defined by the World Health Organization, a certain lack of attention has inhibited progress in fully scaling up treatment and prevention efforts. In addition, despite the fact that the challenges of co-infections have preoccupied clinicians for over two decades, relevant research questions remain unanswered. The following review aims to provide a concise overview of associations between HIV and selected helminthic co-infections concerning aspects of HIV acquisition and transmission, clinical and immunological findings in co-infected individuals, as well as treatment and prevention efforts.

## 1. Introduction

While opportunistic co-infections in HIV-infected individuals have been studied extensively in the past 30 years, helminthic co-infections have not received similar attention, despite being of great clinical significance to millions of people worldwide. Many of the helminthic infections relevant to HIV-infected patients belong to the group of “neglected tropical diseases”, as defined by the World Health Organization. Part of this lack of attention can be attributed to the fact that most patients affected by helminthic infections still live in poverty, without adequate sanitation and in close contact with infectious vectors. In addition, while progress has been made in recent years, additional challenges such as insecurity and weak health systems continue to prevail in the poorest countries, inhibiting progress in scaling up treatment and prevention efforts [1].

For many years, researchers have discussed observations that helminthic infections may increase susceptibility to HIV and that immune activation may have contributed to the extensive spread of HIV in sub-Saharan Africa [2]. In the meantime, immunological studies have revealed humoral and cellular evidence in support of this hypothesis [3,4]. Additionally, cross-sectional clinical studies were able to further support the postulated association between HIV infection and certain helminthic co-infections, particularly for schistosomiasis and lymphatic filariasis [5,6].

The following review focusses on associations between HIV and selected helminthic co-infections concerning aspects of HIV acquisition and transmission, clinical and immunological findings in co-infected individuals, as well as treatment and prevention efforts. While by no means complete, we aim to offer a concise overview on helminthic infections in HIV-positive patients.

## 2. Trematodes

Parasitic flatworms, flukes, or trematodes are a class of unsegmented, soft-bodied invertebrates, which have no body cavity, and no circulatory or respiratory organs. The flat shape of these worms allows oxygen and nutrients to pass through their bodies by diffusion [7]. In human health, the most relevant infections caused by trematodes are schistosomiasis and fascioliasis [8,9]. While schistosomiasis is an infection of tropical and subtropical regions caused by blood flukes (Figure 1), Fascioliasis is a foodborne trematode-infection increasingly prevalent in both the global North and South [10]. Other foodborne trematode-infections include Clonorchiasis, Opisthorchiasis, and Paragonimiasis, which primarily affect domestic and wild animals (zoonotic disease) and cause a variety of pathologies in humans ranging from chronic diarrhea to cholangiocarcinoma (*O. viverrini*, *C. sinensis*) [11]. According to the World Health Organization (WHO), trematode-infections belong to the diverse group of neglected tropical diseases, as they predominantly affect populations living in poverty and with close contact to infectious vectors [12]. In contrast to schistosomiasis, there are very limited reports on the association of foodborne trematode-infections and HIV, which is most likely due to underreporting. 

### 2.1. Schistosomiasis

Infection with schistosomiasis occurs when larval forms are released by freshwater snails and penetrate the human skin during contact with infested water [9,13]. Larvae then transform into adult worms, which live in blood vessels. The eggs of female worms leave the body in the feces or urine. The two main clinical forms of schistosomiasis are intestinal and urogenital schistosomiasis. According to WHO estimations, over 200 Million people were affected by schistosomiasis in 2016, of which >90% live in Africa [14]. Depending on the region of acquisition, intestinal schistosomiasis may be caused by *S. mansoni, S. japonicum, S. mekongi, S. guineensis,* or *S. intercalatum* [13], while urogenital schistosomiasis is mainly caused by *S. haematobium* found in Africa, the Middle East and the Mediterranean island of Corsica [6,13]. Chronic intestinal schistosomiasis usually results in abdominal pain, bloody diarrhea, and wasting. During advanced stages of the disease, liver, and spleen enlargement, as well as ascites are common [13]. The predominant symptom of urogenital schistosomiasis is hematuria and pelvic pain. Advanced cases show fibrosis of the bladder, kidney damage, and genital lesions [13].

#### Urogenital Schistosomiasis and HIV

The great majority of infections with *S. haematobium* occur on the African continent with a substantial epidemiological overlap with regions of high HIV prevalence [4]. In particular, previous studies found young women of rural areas disproportionately affected by both urogenital schistosomiasis and HIV [4]. Increasing evidence from cross-sectional studies suggest an association between genital schistosomiasis and HIV [15,16]. However, the correlation still receives too little attention. 

Urogenital infection with *S. haematobium* is known to cause local chronic inflammation, as the eggs induce a complex cellular and humoral immune response within the infected tissue [4]. If the host is left untreated, eggs are produced for many years [9]. Chronic inflammatory granulomatous lesions can be found in the urinary bladder, and the genitals of men and women are often affected as well. During inflammation, plasma cells, lymphocytes, granulocytes, and macrophages are recruited to the site. These inflammatory cells express CD4+ T-cell receptors [4]. Similar to local lesions caused by HSV-1, HSV-2, or syphilis, the affected tissue has been shown to express more CD4+ T-cell receptors—the primary target of HIV—than healthy tissue [17]. Similar observations were made for infections with *S. mansoni*. Here, a higher density of HIV-1 co-receptors CCR5 and CXCR4 were displayed on monocytes and CD4+ T-cells of infected individuals [18]. After penetration through inflamed genital mucosa, the authors hypothesize that higher receptor expression facilitates rapid binding of HIV in patients with schistosomiasis. Thus, chronic infection with schistosomiasis is a risk factor for HIV acquisition through genital lesions and local mucosal inflammation with overexpression of HIV co-receptors.

Schistosomiasis-associated pathological alterations in the genitalia of women have been defined as a special entity termed female genital schistosomiasis (FGS), which is most commonly caused by *S. haematobium*, less often by *S. mansoni* [6]. Approximately, 20 million women of reproductive age are affected by FGS worldwide, with the prevalence ranging from 33–75% [19]. In many countries, FGS is the most common cause of genital lesions in women [15,20]. The diagnosis of FGS is particularly tricky, as it may be mistaken for cervical cancer or sexually transmitted diseases and requires a gynecological examination by a trained health care professional [19]. Cross-sectional studies from Zimbabwe and Tanzania demonstrated a three- and four-fold increased risk of HIV infections in women with FGS [21,22]. Similarly, a clinical cohort study from Zambia was able to show that schistosomiasis was associated with an increased risk of HIV transmission from both sexes, and an increased risk of HIV acquisition in women [17]. The same study also demonstrated an increased risk of HIV disease progression and death in HIV-infected women. FGS in HIV-infected women is also of particular concern, as there is reason to suspect a higher probability of HIV transmission to sexual partners, as well as from mother to child due to increased virus-carrying inflammatory cells in vaginal fluids [23,24]. Similarly, urogenital schistosomiasis in men has been hypothesized to increase the risk of HIV transmission through increased shedding of HIV RNA in semen [25]. Postmortem studies from the 1970s were able to show schistosomiasis-induced chronic inflammation of the prostate and seminal bladder [26]. Furthermore, leucocytospermia and high seminal levels of inflammatory cytokines as seen in men with urogenital schistosomiasis actually normalize after treatment of schistosomiasis with praziquantel [27]. A recent prospective clinical study assessed the effect of praziquantel treatment on HIV-1 RNA load in blood plasma and semen of HIV-infected men from Zimbabwe co-infected with urogenital schistosomiasis [28]. The study demonstrated a decline of seminal HIV-1 RNA load following praziquantel treatment. Thus, a reduction of the risk of HIV transmission through mass administration of praziquantel can be assumed [29].

Strategies to control schistosomiasis have been employed, including mass chemotherapy, improvements to sanitation, modification of the environment, and the use of molluscicides [30]. Praziquantel, the drug of choice in the treatment of schistosomiasis, is safe, effective, and inexpensive [30]. However, it is less active against juvenile than mature parasites [30]. WHO recommends periodic administration of preventive chemotherapy with praziquantel to reduce morbidity associated with schistosomiasis in endemic areas [31]. In 2017, 98.7 million people received praziquantel for schistosomiasis, which corresponds to 44.9% of the target numbers [31]. While the global treatment coverage for school-aged children increased to 68%, only 16,9% of adults were treated. Taking current evidence into account, mass treatment with praziquantel would most likely also be an innovative additional measure for HIV prevention in regions endemic for schistosomiasis. Unfortunately, the main limitation to increasing treatment coverage is the absence of praziquantel donations, as outlined by the WHO [31]. Currently, praziquantel is available free of charge to high-disease burden countries in sub-Saharan Africa, through a donation from Merck Serono. Additional praziquantel is available to selected countries with funding from the Department for International Development (DfID) and the USAID [14]. 

## 3. Nematodes

### 3.1. Filarial Nematode Infections 

Filariasis is caused by infection with nematodes of the Filarioidea type. Common diseases caused by these worms include infections with *Wuchereria bancrofti*, *Loa loa*, and *Onchocerca volvulus,* among others. While *Wuchereria bancrofti* is spread through Mosquitoes and causes lymphatic filariasis, *Loa* and *Onchocerca volvulus* belong to the so-called subcutaneous filarial diseases and are spread through the bite of horse flies and black flies, respectively [32]. Both onchocerciasis and lymphatic filariasis are considered neglected tropical diseases according to WHO.

Lymphatic filariasis is caused by *Wuchereria bancrofti* or *Brugia* species and affects approximately 120 million people in subtropical and tropical regions of Africa and South-East Asia [33]. In high prevalence areas such as Southwest Tanzania, almost 25% of the population have circulating filarial antigen [34]. This chronic helminth infection, in which adult worms reside in the lymphatic system for many years, may result in severe, disfiguring lymphedema (“elephantiasis”). 

#### 3.1.1. Lymphatic Filariasis and HIV

The geographical distribution of lymphatic filariasis correlates with regions of high HIV prevalence (Figure 2). A recent prospective cohort study from Tanzania which enrolled over 4000 households with approximately 18,000 study participants found a significantly higher HIV incidence in lymphatic filariasis-positive participants than in lymphatic filariasis-negative participants [35]. For instance, a three-fold increase in HIV incidence was shown in the adolescent group. When controlled for other risk factors such as sexual behavior and socioeconomic factors, lymphatic filariasis remained an independent and significantly relevant risk factor for HIV infection, as reported by the authors [35]. So far, the cause for increased HIV acquisition in patients with lymphatic filariasis is unknown. However, systemic immune response activation through helminth infections may play a role in the facilitation of early viral dissemination. For instance, patients with lymphedema due to lymphatic filariasis show an increased pro-inflammatory Th17 profile [36]. Furthermore, changes in the cytokine profile of peripheral blood mononuclear cells in patients with lymphatic filariasis are associated with an increased HIV-susceptibility in vitro [37]. Currently, immunological, as well as interventional treatment studies are needed to follow-up on findings of associations between HIV and lymphatic filariasis as this may bring to light an important instrument to further reduce HIV transmission in endemic regions.

#### 3.1.2. “River Blindness” and HIV

The filarial worm *Onchocerca volvulus* is transmitted from person to person by infected black flies and causes the so-called “river blindness”. Main symptoms are severe itching and visual impairment including permanent blindness. Onchocerciasis is the most important cause of infectious blindness worldwide [38]. According to the Global Burden of Disease Study, 20.9 million people were infected with *O. volvulus* in 2017, of which 99% lived in African countries and over 1 million had visual loss [38]. Efforts to reduce onchocerciasis include vector control, as well as mass distribution of the drug ivermectin, according to the African Programme for Onchocerciasis Control (POC) [39].

In general, epidemiological mapping of *O. volvulus* shows a substantial overlap with regions of high HIV prevalence (Figure 3). Uganda, for instance, has been burdened by both infections and substantial research on potential associations of the two immunocompromising diseases has been done by Ugandan researchers [40]. For example, a study on the humoral immune response was able to show that HIV-infected patients with onchocerciasis had an impaired antibody response to *O. volvulus* antigens [41]. Furthermore, compared to patients without HIV co-infection, these patients tended to lose their reactivity to these antigens over time due to immune response abnormalities attributed to HIV [41]. T-cell responses in co-infected patients were studied by Ugandan researchers as well. Here, the cellular immune response in co-infected patients suggested a lack of specific reaction to *O. volvulus*, again attributed to HIV [42]. Additionally, the authors found an impaired IL-4 and IL-5 production, as well as a lack of interferon-gamma response to antigen stimulation. In summary, there is evidence for a reduced humoral and cellular immune response in HIV-infected patients with onchocerciasis as compared to HIV-negative patients. However, the studies mentioned here were done before broad access to antiretroviral treatment and so far, there is no data available on the clinical course of onchocerciasis specifically in HIV-infected patients.

### 3.2. Strongyloides Stercoralis

*Strongyloides stercoralis* is a soil-transmitted nematode helminth, which is distributed globally with most cases found in warm moist areas of the tropics and subtropics [43]. Acute infection with *Strongyloides* results in cutaneous (ground itch) or pulmonary symptoms (dry cough, wheeze, dyspnea, Loeffler’s syndrome), while the larvae migrate through the body [43]. Most patients with chronic infections are asymptomatic or present with cutaneous (*Larva currens*, chronic urticaria), pulmonary or abdominal symptoms (diarrhea, anorexia, vomiting, epigastric pain). In contrast to other intestinal parasites, *Strongyloides* can re-infect the host by developing the infective stage (*filarifom larvae*) inside the human intestine. The larvae subsequently penetrate the intestinal wall or the perianal skin and restart migrating through the body (*autoinfection*). This peculiarity enables chronic infection which can last for decades [44].

#### Strongyloides and HIV

In immunosuppressed patients infected with *Strongyloides* the so-called Strongyloidiasis Hyperinfection Syndrome (SHS) may occur [45]. In that case, immunosuppression leads to accelerated autoinfection with uncontrolled multiplication and possible dissemination of the larvae (disseminated strongyloidiasis, DS) to other organs outside the normal intestinal-pulmonary-lifecycle (e.g., brain, liver, heart). This, along with an often-accompanying gram-negative sepsis due to the translocation of gut bacteria, leads to a life-threatening situation with a high mortality (>60%) [46]. Main risk factors for SHS are treatment with corticosteroids (e.g., for chronic obstructive pulmonary disease or lymphoma), transplant patients and infection with the human T-lymphotropic virus type 1 (HTLV-1).

HIV is a risk factor for *Strongyloides* infection, and co-infection is common in endemic regions. In a meta-analysis, the overall prevalence of strongyloidiasis in HIV-infected patients was 10% with a 2-fold higher risk of infection compared to HIV-negative controls [47]. As serological testing can be unreliable in severely immunosuppressed patients, some authors advocate for empiric treatment of strongyloidiasis with ivermectin in HIV-infected patients from endemic countries, especially before initiating corticosteroid therapy to reduce the risk of SHS [48]. Others have suggested a systematic screening for *Strongyloides* infection with both serology and stool agar culture for all HIV-infected patients coming from endemic countries [49]. 

Despite the increased risk of *Strongyloides stercoralis* infection in HIV-infected patients and the fact that an impaired cell-mediated immunity is a risk factor for SHS and DS, these severe forms are rarely reported in HIV-infected patients [50]. One explanation could be the Th1 to Th2 cytokine shift with elevated levels of IL-4, IL-5, and consequently IgE and eosinophils in HIV infection leading to conserved anti-helminthic immunity in AIDS-patients [50]. Another reason could be an impaired direct development of autoinfective filarifom *Strongyloides* larvae in the gut and a favored indirect development into non-infective free-living adults observed in individuals immunosuppressed by advanced HIV infection [51]. However, the lack of reports on severe cases of strongyloidiasis in patients with HIV could also be attributed to under diagnosis [52].

The fatality rate in HIV-infected patients with SHS is very high [52]. Furthermore, several cases of unmasking and paradoxical Immune Reconstitution Inflammatory Syndrome (IRIS) due to *Strongyloides stercoralis* occurring 19–150 days after antiretroviral treatment (ART) initiation with the clinical picture of SHS or DS have been reported [53]. Clinical deterioration with immune reconstitution is probably related to a positive correlation of CD4^+^ T-cell count and a favored direct development of infective filariform larvae in the gut [54]. Prolonged dual anti-helminthic treatment with ivermectin and albendazole and temporarily stopping ART is recommended in these patients [54].

In conclusion, clinicians should be aware of the potentially fatal co-infection with Strongyloides in immunosuppressed HIV-infected patients from endemic countries. Screening and subsequent treatment for strongyloidiasis before initiating ART and/or corticosteroids in high risk patients is recommended [52,54].

## 4. Cestoda (Tapeworms)

Echinococcosis and taeniasis are helminthic zoonotic diseases caused by cestoda (tapeworms). A segmented body structure is typical for these parasites, which consist of single limbs (proglottides), a head (scolex) with suction cups, and a head-near growth zone. Tapeworms do not have a digestive tract, but feed by diffusion. 

### 4.1. Human Echinococcosis

Human Echinococcosis is caused by tapeworms of the genus Echinococcus. Epidemiologically relevant are *E. granulosus* (“dog tapeworm”) with multiple subspecies (*E. canadensis*, *E. ortleppi*, *E. equinus*), and *E. multilocularis* (“fox tapeworm”) [55]. While *E. granulosus* is distributed worldwide, *E. multilocularis* can only be found in the Northern hemisphere (Figure 4) [56]. Furthermore, sporadically occurring neotropical species include *E. vogeli* and *E. oligarthus* in Central and South America [57].

The life cycle of the different species is similar; However, definitive and intermediate host species differ. Adult Echinococcus spp. colonize the small intestine of their definitive hosts (*E. granulosus*: dogs and other canine animals; *E. multilocularis*: mainly foxes, but also dogs, cats, coyotes, wolves). In the intestine, eggs are released from the gravid proglottids and are distributed into the environment with feces. Intermediate hosts (*E. granulosus*: sheep, goats, pigs, cows, horses, camels; *E. multilocularis*: rodents) ingest infectious eggs and develop organ cysts (liver, lung, kidney, brain, muscle). As soon as an intermediate host is eaten by a final host, the infection cycle is closed. Humans are so-called “dead-end hosts”. 

#### Human Echinococcosis and HIV

Reports on co-infections with HIV and Echinococcus spp. are limited, although they have an overlapping geographical distribution (China, India, Brazil, Central and Southern African countries) [58,59]. Accordingly, little is known about the interaction of both infectious agents. Most previous studies and reports have focused on infections with *E. granulosus*, while information on co-infections with other species is scarce [60].

Previous observations have come up with the following associations between human echinococcosis and HIV: Echinococcosis usually leads to a TH2 (T helper)-dominated immunomodulation, and suppression of the CD4+ T-cell population, as is typical in HIV infection [61]. Since CD4+ T-cells are usually able to limit cyst growth to a certain extent, complicated clinical forms of echinococcosis such as neuronal involvement, multifocal cysts, and racemic forms are observed more often in HIV-infected individuals with impaired immune function [62,63,64]. However, so far no association has been found between a low CD4+ T-cell count and the probability of acquiring human echinococcosis [59]. 

Curative treatment including cystectomy and specific anthelminthic treatment can significantly increase the CD4+ T-cell count in HIV co-infected patients [61]. The risk of developing an IRIS following the treatment of echinococcosis in HIV co-infected patients is currently unknown. It should be noted that the serological diagnosis of echinococcosis in HIV-infected patients may be challenging. Due to reduced antibody responses, serological sensitivity is particularly poor in HIV-infected individuals [64,65,66]. A negative result does not rule out echinococcosis, and seroprevalence studies should be interpreted with great caution, as there is a risk of underestimation [59].

### 4.2. Taeniasis

In general, taeniasis is caused by either *T. solium* (pork tapeworm) or *T. saginata* (beef pork worm), which are commonly transmitted to humans through oral egg intake via raw or poorly cooked infected meat. Humans are the only definitive hosts. In the human intestine, the cysticercus develops into an adult worm, which can become several meters long and survive for many years. Worm proglottids are excreted with feces and the infectious eggs are released into the environment. Intermediate hosts ingest eggs and develop a cystic disease. Upon infection of a human through meat, the circle is closed.

In contrast to infections with *T. saginata*, infection with *T. solium* can result in severe cystic disease of the muscles, central nervous system (CNS), and other organs. The clinical picture is referred to as cysticercosis (CC) or neurocysticercosis (NCC). In sub-Saharan Africa, CC is an emerging public health problem especially among subsistence farming communities (Figure 5) [67,68]. Here, pigs roam freely, and meat inspection is uncommon [59]. Further risk factors for tapeworm infections include poor sanitation, political and economic instabilities, and close contact to patients suffering from taeniasis [67]. 

#### T. Solium and HIV

In HIV-infected patients, CC and NCC are relevant co-infections [67]. In South Africa, up to 27% of CNS lesions in HIV-infected patients were due to NCC [69]. So far, little is known about how the clinical picture of NCC is modified by HIV; however, the most frequent presentation of NCC in HIV co-infection seems to be a multi-cystic manifestation [67,70]. Besides an impact on the clinical presentation of taeniasis, HIV may also impact the serological diagnostics [71]. Previous observations support the hypothesis that HIV-infected patients with higher CD4+ T-cell counts are more likely to develop symptomatic NCC needing treatment, as compared to patients with advanced HIV and lower CD4+ counts [67]. However, this observation was not supported in other studies [59]. In addition, activation of latent NCC in patients initiating ART in the context of an IRIS, should be mentioned here as this can have potentially deleterious effects for the affected individual [67].

## 5. Conclusions

The associations between HIV and helminthic infections are as heterogeneous as the variety of species and subspecies belonging to this group of parasites. Even though the challenges of co-infections have preoccupied clinicians for over two decades, relevant research questions remain unanswered. Thus, well designed, controlled intervention studies which aim to provide definitive information on the immunological aspects of interactions between HIV and helminths, as well as on optimal type and timing of de-worming in relation to HIV infection, are urgently needed. As the authors J. Downs and D. Fitzgerald have recently put it in The Lancet: “Don’t 2 billion people living with these neglected tropical diseases, who might be at increased risk of HIV infection, deserve such attention?” [2].

## Figures and Tables

**Figure 1 tropicalmed-04-00065-f001:**
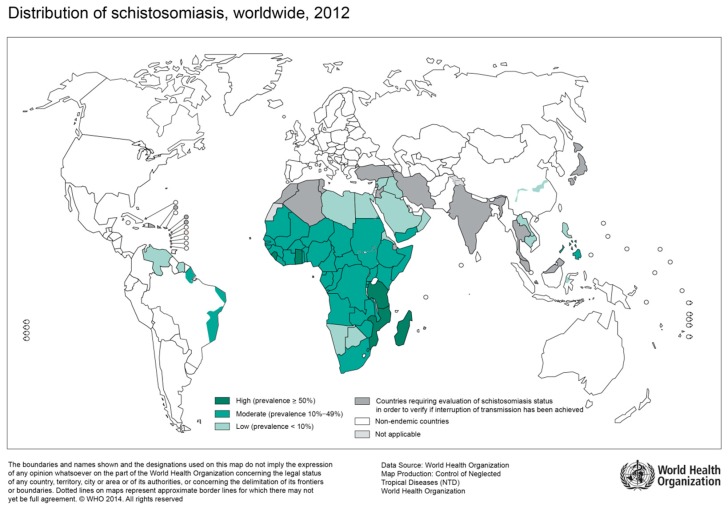
Global distribution of countries where human schistosomiasis is transmitted [9]. (Source: WHO, website: https://www.who.int/schistosomiasis/Schistosomiasis_2012-01.png?ua=1)

**Figure 2 tropicalmed-04-00065-f002:**
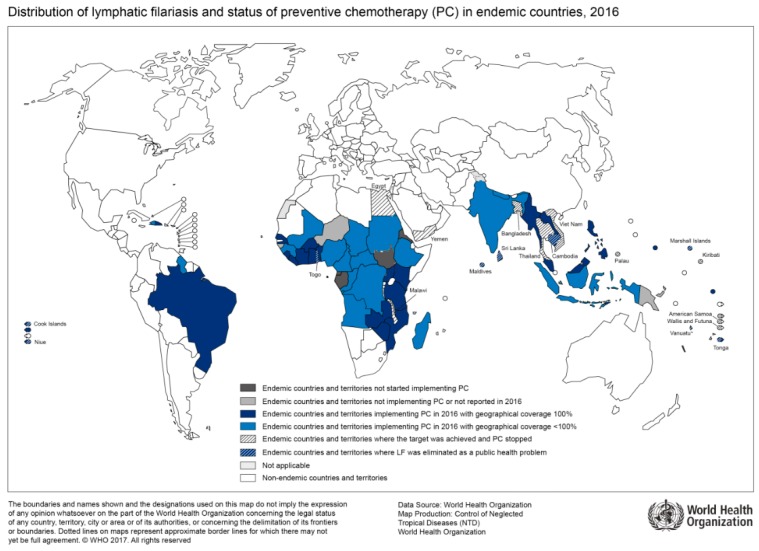
Countries endemic for lymphatic filariasis and status of mass drug administration, 2016 (Source: WHO, website: http://gamapserver.who.int/mapLibrary/Files/Maps/LF_2016.png).

**Figure 3 tropicalmed-04-00065-f003:**
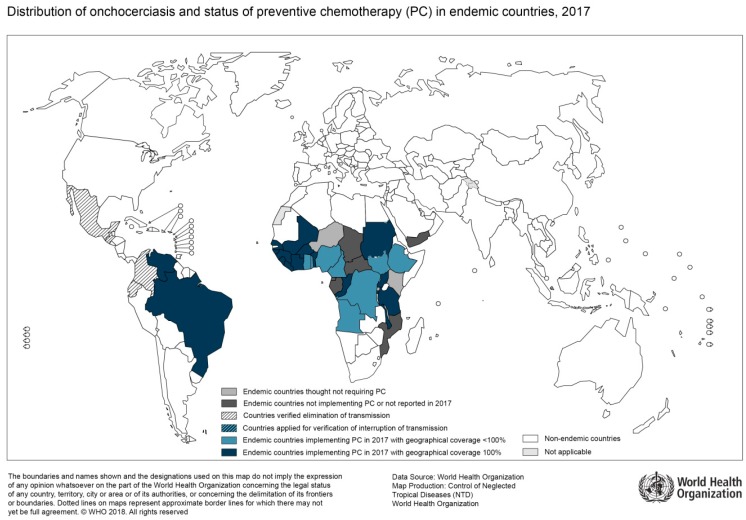
Distribution of Onchocerciasis, 2017 (Source: WHO; website: http://gamapserver.who.int/mapLibrary/Files/Maps/Onchocerciasis_2017.png).

**Figure 4 tropicalmed-04-00065-f004:**
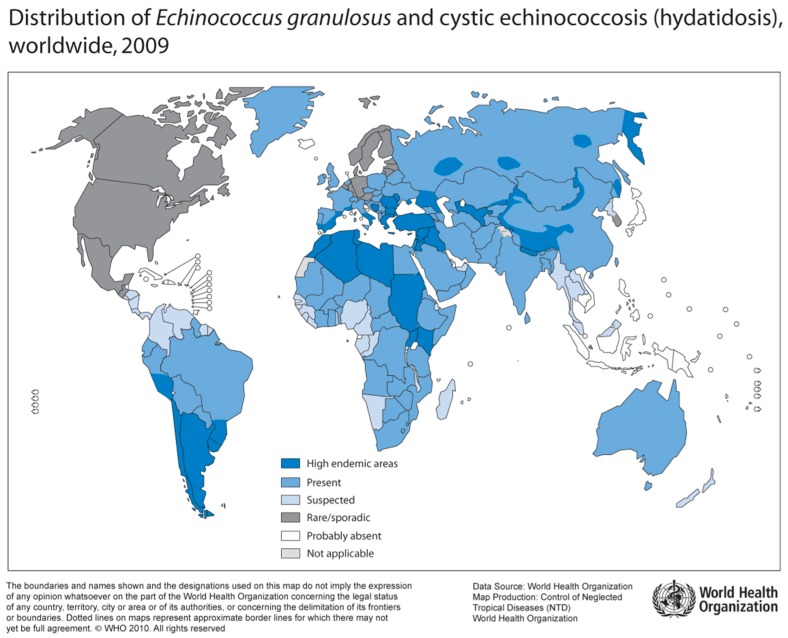
Distribution of *E. granulosus* and cystic hydatosis, 2009 (Source: WHO, website: http://gamapserver.who.int/mapLibrary/Files/Maps/Global_echinococcosis_2009.png).

**Figure 5 tropicalmed-04-00065-f005:**
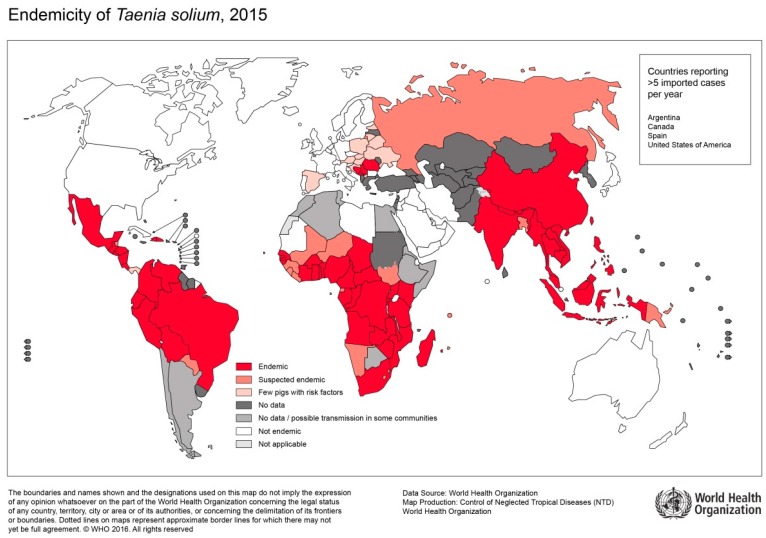
Endemicity of T.solium, 2015 (Sorce: WHO, website: https://www.who.int/taeniasis/Endemicity_Taenia_Solium_2015.jpg?ua=1).

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
