# Peer review of "Schistosoma and Other Relevant Helminth Infections in HIV-Positive Individuals—An Overview"

_tropicalmed, 2019, doi:10.3390/tropicalmed4020065_

Round 1
Reviewer 1 Report
The review article by Braun et al on helminth infections in relation to HIV infections is very significant for two reasons. 1. Helminth infections in HIV patients and the consequences are poorly understood. 2. Whether and how helminth infections can play a role in HIV related complications is not researched. With this, this review article is of significance to the researchers and the public, equally. Nevertheless, some of the critiques are as follows, and addressing of those may significantly enhance the value of the review article submission.
Major comments:
It is worth to put Fascioliasis prevalence on the map to show the significance, either by different a color or by a different map. Line 50-51.
Line 87: Please clarify in relation to CD4 T cells and non-T cells. It is unclear with the word 'CD4+ T cell receptors'. what does it mean?
Lymphatic filariasis-Stating stats of occurrence/prevalence is encouraged, may be included in the conclusion/discussion section, if preferred.
Authors may have to clarify what is an egg or chicken. If both, please state so, clearly. or Rephrase sentence like 'both HIV and the parasitic infections coordinate. Regards to Line 206.
Line 296. 'relevant questions remain unanswered'. What are those? or what are authors referring to?
Line 298: There are many parasites with continuous exposure and de-worming is temporary. How do you postulate to control the mentioned parasites in the long run, if not all?
How do you separate the HIV infection-induced CD4 T cell deficiency with the occurrence of parasitic infections and parasite infection associated susceptibility to HIV? It is worth to discuss in the conclusion section.
Minor comments:
Line 47-48. Rephrase.
Line 72. English-rephrase.
Line 124 requires a reference.
Line 127-donations-explain or clarify or reword. 'Donation' is perceived differently, in general.
Line 148, strikingly, especially-please rephrase the sentence.
Line 154, profile spell check
Line 174/176 grammar
Line 180-correct abbreviation of Interferon gamma
Line 182-grammar
Line 189 grammar
Line 194 grammar
Line 197 rephrase
Author Response
Point-by-point reply to reviewers
We would like to express our sincere gratitude to the reviewers for the precise and timely review of our manuscript. The comments provided to us have been very helpful.
Unfortunately, there seems to have been a shift of the lines referred to by reviewer no. 1, so that we were not able to identify all suggestions under “minor comments”. Perhaps this can be clarified further by the editor. Furthermore, concerning the suggestions made for the maps: these were adjusted in a harmonized way by choosing WHO maps only. We hope this is in-line with the reviewers’ preferences. Kindly find the reply to reviewers below.
Reviewer no. 1
The review article by Braun et al on helminth infections in relation to HIV infections is very significant for two reasons. 1. Helminth infections in HIV patients and the consequences are poorly understood. 2. Whether and how helminth infections can play a role in HIV related complications is not researched. With this, this review article is of significance to the researchers and the public, equally. Nevertheless, some of the critiques are as follows, and addressing of those may significantly enhance the value of the review article submission.
Major comments:
Line 87: Please clarify in relation to CD4 T cells and non-T cells. It is unclear with the word 'CD4+ T cell receptors'. what does it mean?
Response: thank you for this comment. The sentence referred to has been changed to the following for clarification: “During inflammation, plasma cells, lymphocytes, granulocytes, and macrophages are recruited to the site. These inflammatory cells express CD4+ T-cell receptors.” The correct reference was added as well. (line 88-90)
Lymphatic filariasis-Stating stats of occurrence/prevalence is encouraged, may be included in the conclusion/discussion section, if preferred.
Response: Following the sentence “Lymphatic filariasis is caused by Wuchereria bancrofti or Brugia species and affects approximately 120 million people in subtropical and tropical regions of Africa and South-East Asia”, we added a sentence on prevalence: “In high prevalence areas such as Southwest Tanzania, almost 25% of the population have circulating filarial antigen.”(line 148-150)
Authors may have to clarify what is an egg or chicken. If both, please state so, clearly. or Rephrase sentence like 'both HIV and the parasitic infections coordinate. Regards to Line 206.
Response: Here, it is unclear to us what was meant by the reviewer. Unfortunately, the comment made does not refer to line 206, and the sentence fragment cited cannot be found in the text.
Line 296. 'relevant questions remain unanswered'. What are those? or what are authors referring to?
Response: thank you for this comment. In our conclusion, we suggest to further study immunological aspects of interactions between HIV and helminths, as well as the optimal type and timing of de-worming in relation to HIV infection. These are some of the questions that are to-date unanswered but of high clinical relevance to populations affected.
The section reads as follows: “Despite the fact that the challenges of co-infections have preoccupied clinicians for over two decades, relevant research questions remain unanswered. Thus, well designed, controlled intervention studies which aim to provide definitive information on the immunological aspects of interactions between HIV and helminths, as well as on optimal type and timing of de-worming in relation to HIV infection, are urgently needed.”(line 316 onwards)
Line 298: There are many parasites with continuous exposure and de-worming is temporary. How do you postulate to control the mentioned parasites in the long run, if not all?
Response: This is a very good question, thank you. According to the WHO framework “Global Progress Towards Elimination” mass drug administration (MDA) is key to stopping the spread of many parasites, for instance lymphatic filariasis. Yes, even though de-worming once is temporary, repeated de-worming can be successful: By following WHO recommendations on repeated de-worming (MDA), so far China and the Republic of Korea were able to eliminate lymphatic filariasis as a public health problem. Further 18 countries have completed interventions and are conducting surveillance to validate elimination. The main challenge for many other countries is to achieve 100% geographical coverage of MDA. These countries are in need of continuous logistical and financial support by the Global Health Community. We added the following introductory sentence to the paragraph referred to: “Strategies to control Schistosomiasis have been employed, including mass chemotherapy, improvements to sanitation, modification of the environment, and the use of molluscicides.” (lines 122-123)
How do you separate the HIV infection-induced CD4 T cell deficiency with the occurrence of parasitic infections and parasite infection associated susceptibility to HIV? It is worth to discuss in the conclusion section.
Response: Thank you for this valuable comment. It was our aim to address the issue of associated susceptibility to HIV, as well as immunological modulations of parasitic infections in HIV-positive patients in the specific sections of our manuscript, rather than further summing this up in the conclusion. We chose this approach due to the fact that the parasites are so heterogeneous especially concerning immunological mechanisms and HIV susceptibility. Thus, we considered it more concise for readers to focus on these aspects “one worm at a time”. Nevertheless, we fully agree with the reviewer that the question raised here is of great importance, and hope to have taken this into account in the sections of the manuscript.
Minor comments:
Line 47-48. Rephrase.
Response: The sentence was rephrased to: “The flat shape of these worms allows oxygen and nutrients to pass through their bodies by diffusion.”(lines 49-50)
Line 72. English-rephrase.
Response: The sentence was rephrased to: “During advanced stages of the disease, liver and spleen enlargement, as well as ascites are common.”(lines 74-75)
Line 124 requires a reference.
Response: The reference was added.
Line 127-donations-explain or clarify or reword. 'Donation' is perceived differently, in general.
Response: The following information was added in order to clarify this aspect: “Currently, praziquantel is available free of charge to high-disease burden countries in sub-Saharan Africa, through a donation from Merck Serono. Additional praziquantel is available to selected countries with funding from the Department for International Development (DfID) and the USAID.”(lines 133-136)
Line 148, strikingly, especially-please rephrase the sentence.
Response: The sentence was rephrased to: “For instance, a three-fold increase in HIV incidence was shown in the adolescent group.”(lines 156-157)
Line 154, profile spell check
Response: the error was corrected.(line 163)
Line 174/176 grammar
Response: No changes made, as it is unclear to which sentence the reviewer is referring to exactly.
Line 180-correct abbreviation of Interferon gamma
Response: The abbreviation was taken changed to Interferon-gamma. (line 189)
Line 182-grammar
Response: No changes made, as it is unclear to which sentence the reviewer is referring to exactly.
Line 189 grammar
Response: No changes made, as it is unclear to what the reviewer is referring to exactly. The line given by the reviewer is a heading.
Response: No changes made, as it is unclear to which sentence the reviewer is referring to exactly.
Line 197 rephrase
Response: No changes made, as it is unclear to which sentence the reviewer is referring to exactly.
Reviewer 2 Report
This review focusses on the association of Schistosomiasis and other relevant helminth infections in HIV-positive individuals, highlighting, in some cases, the immunological mechanisms of interactions between HIV and helminths.
Limitation: The association between S. mansoni and HIV was mentioned sparingly, perhaps due to the dearth or lack of experiment-based evidence in the area. The authors should make as clear as possible what are the relevant research questions remain unanswered.
Strength: The review covers a nice and concise breadth of the topic, and can trigger new research on schistosome/HIV-positive individuals interactions.
From line 46 to 49: the reference is needed.
59: Schistosomiasis map does not show Corsican strain (mixed S. haematobium and S. bovis)
127: Please add information about why mass treatment based just only in praziquantel is not sufficient to interrupt the life cycle.
130-135: the reference is needed
136: Filariasis map should be organized in the same way that schistosomiasis
168: Onchocerciasis map should be organized in the same way that schistosomiasis
234: Please add information about tapeworm life cycle
245: Echinococcosis map should be organized in the same way that schistosomiasis
280: Please add to the map information about Taenia saginata and the Taenia solium map should be organized in the same way that schistosomiasis
Author Response
Point-by-point reply to reviewers
We would like to express our sincere gratitude to the reviewers for the precise and timely review of our manuscript. The comments provided to us have been very helpful.
Unfortunately, there seems to have been a shift of the lines referred to by reviewer no. 1, so that we were not able to identify all suggestions under “minor comments”. Perhaps this can be clarified further by the editor. Furthermore, concerning the suggestions made for the maps: these were adjusted in a harmonized way by choosing WHO maps only. We hope this is in-line with the reviewers’ preferences. Kindly find the reply to reviewers below.
Reviewer no. 2:
This review focusses on the association of Schistosomiasis and other relevant helminth infections in HIV-positive individuals, highlighting, in some cases, the immunological mechanisms of interactions between HIV and helminths.
Limitation: The association between S. mansoni and HIV was mentioned sparingly, perhaps due to the dearth or lack of experiment-based evidence in the area. The authors should make as clear as possible what are the relevant research questions remain unanswered.
Strength: The review covers a nice and concise breadth of the topic, and can trigger new research on schistosome/HIV-positive individuals interactions.
From line 46 to 49: the reference is needed.
Response: The references were added accordingly.
59: Schistosomiasis map does not show Corsican strain (mixed S. haematobium and S. bovis)
Response: Thank you for pointing this out. We were unable to find a more complete map, unfortunately. We now changed the initial map to the most recent WHO map. (line 62)
127: Please add information about why mass treatment based just only in praziquantel is not sufficient to interrupt the life cycle.
Response: Thank you for this comment. The following information was added to the section referred to: “Strategies to control Schistosomiasis have been employed, including mass chemotherapy, improvements to sanitation, modification of the environment, and the use of molluscicides. Mass drug administration is done with praziquantel, the drug of choice in the treatment of Schistosomiasis, which is safe, effective, and inexpensive. However, it is less active against juvenile than mature parasites.”(line 125)
130-135: the reference is needed
Response: The reference was added.
234: Please add information about tapeworm life cycle
Response:
The following information was added to the section on echinococcosis:
“The life cycle of the different species is similar; However, definitive and intermediate host species differ. Adult Echinococcus spp. colonize the small intestine of their definitive hosts (E. granulosus: dogs and other canine animals; E. multilocularis: mainly foxes, but also dogs, cats, coyotes, wolves). In the intestine, eggs are released from the gravid proglottids and are distributed into the environment with feces. Intermediate hosts (E. granulosus: sheep, goats, pigs, cows, horses, camels; E. multilocularis: rodents) ingest infectious eggs and develop organ cysts (liver, lung, kidney, brain, muscle). As soon as an intermediate host is eaten by a final host, the infection cycle is closed. Humans are so-called ‘dead end hosts’.”(line 258 onwards)
The following information was added to the section on Taeniasis:
“Humans are the only definitive hosts. In the human intestine, the cysticercus develops into an adult worm, which can become several meters long and survive for many years. Worm proglottids are excreted with feces and the infectious eggs are released into the environment. Intermediate hosts ingest eggs and develop a cystic disease. Upon infection of a human through meat, the circle is closed.”(line 290 onwards)
Reviewers 1 & 2 Maps:
It is worth to put Fascioliasis prevalence on the map to show the significance, either by different a color or by a different map.
136: Filariasis map should be organized in the same way that schistosomiasis
168: Onchocerciasis map should be organized in the same way that schistosomiasis
245: Echinococcosis map should be organized in the same way that schistosomiasis
280: Please add to the map information about Taenia saginata and the Taenia solium map should be organized in the same way that schistosomiasis
Response: The maps used in this manuscript are provided by the WHO Global Health Observatory Map Gallery. In the section on Schistosomiasis, we initially used a map from the Lancet. However, following the comments made by the reviewers we changed this to a WHO map for harmonization. There is, unfortunately, no WHO map on Taenia saginata or Fascioliasis available.
Round 2
Reviewer 1 Report
Authors have addressed the important points and the revised version is suitable for publication.